# Multicenter validation of a machine learning phase space electro-mechanical pulse wave analysis to predict elevated left ventricular end diastolic pressure at the point-of-care

Sanjeev P. Bhavnani[1]*, Rola Khedraki[2], Travis J. Cohoon[1], Frederick J. Meine, III[3], Thomas D. Stuckey[4], Thomas McMinn[5], Jeremiah P. Depta[6], Brett Bennett[7], Thomas McGarry[8], William Carroll[9], David Suh[10], John A. Steuter[11], Michael Roberts[12], Horace R. Gillins[13], Ian Shadforth[13], Emmanuel Lange[14,15], Abhinav Doomra[14,15], Mohammad Firouzi[14,15], Farhad Fathieh[14,15], Timothy Burton[14,15], Ali Khosousi[14,15], Shyam Ramchandani[14,15], William E. Sanders, Jr.[13], Frank Smart[16]

1 Division of Cardiovascular Medicine, Healthcare Innovation & Practice Transformation Laboratory, Scripps Clinic, San Diego, California, United States of America, 2 Division of Cardiology, Section Advanced Heart Failure, Scripps Clinic, San Diego, California, United States of America, 3 Novant Health New Hanover Regional Medical Center, Wilmington, North Carolina, United States of America, 4 Cone Health Heart and Vascular Center, Greensboro, North Carolina, United States of America, 5 Austin Heart, Austin, Texas, United States of America, 6 Rochester General Hospital, Rochester, New York, United States of America, 7 Jackson Heart Clinic, Jackson, Mississippi, United States of America, 8 Oklahoma Heart Hospital, Oklahoma City, Oklahoma, United States of America, 9 Cardiology Associates of North Mississippi, Tupelo, Mississippi, United States of America, 10 Atlanta Heart Specialists, Atlanta, Georgia, United States of America, 11 Bryan Heart, Lincoln, Nebraska, United States of America, 12 Lexington Medical Center, West Columbia, South Carolina, United States of America, 13 CorVista Health, Inc., Washington, DC, United States of America, 14 CorVista Health, Toronto, Ontario, Canada, 15 Analytics For Life Inc., d.b.a CorVista Health, Toronto, Canada, 16 LSU Health Science Center, New Orleans, Louisiana, United States of America

* Bhavnani.sanjeev@gmail.com

## Abstract

### Background

Phase space is a mechanical systems approach and large-scale data representation of an object in 3-dimensional space. Whether such techniques can be applied to predict left ventricular pressures non-invasively and at the point-of-care is unknown.

### Objective

This study prospectively validated a phase space machine-learned approach based on a novel electro-mechanical pulse wave method of data collection through orthogonal voltage gradient (OVG) and photoplethysmography (PPG) for the prediction of elevated left ventricular end diastolic pressure (LVEDP).

### Methods

Consecutive outpatients across 15 US-based healthcare centers with symptoms suggestive of coronary artery disease were enrolled at the time of elective cardiac catheterization and underwent OVG and PPG data acquisition immediately prior to angiography with signals

**Data Availability Statement:** All relevant data are within the paper and its Supporting information files.

**Funding:** The study sponsor (CorVista Health; Washington, DC & Toronto, Canada - corvista. com) was involved in the study design, data analysis, and interpretation of the results. The corresponding author, Sanjeev Bhavnani, had full access to all the data in the study, data analysis, and had final responsibility for the decision to submit for publication.

**Competing interests:** I have read the journal's policy and authors of this manuscript have the following competing interests. Sanjeev Bhavnani MD is a scientific advisor to Corvista Health and Blumio; consultant to Bristol Meyers Squibb, Pfizer, and Infineon Semiconductor; data safety monitoring board chair at Proteus Digital; has received research support from Scripps Clinic and the Qualcomm Foundation and is member of the healthcare innovation advisory boards at the American College of Cardiology, American Society of Echocardiography, and BIOCOM (all non-profit institutions with all positions voluntary). Jeremiah P. Depta MD reports consulting fees from Edwards Lifesciences LLC, Boston Scientific, V wave Medical Ltd and Abbot. Brett Bennett MD reports payment or honoraria for lecture from Philips. Horace R. Gillins BS, Ian Shadforth EngD, Emmanuel Lange, Abhinav Doomra MScAC, Mohammad Firouzi MSc, Farhad Fathieh PhD, Timothy Burton BComp, Ali Khosousi PhD, Shyam Ramchandani PhD and William E. Sanders Jr. MD report employment by CorVista Health, and stock options in the same. Frank Smart MD reports grants or contracts from Abbot (GUIDE HF clinical trial), NIH / Ohio State (DCM genetic study), Duke Clinical Research (Transform HF), CorVista Health (Pulmonary Hypertension clinical trial), and participation on a Data Safety Monitoring Board or Advisory Board (Abbott Medical; GUIDE-HF Steering committee). All other authors report no disclosures. This does not alter our adherence to PLOS ONE policies on sharing data and materials.

**Abbreviations:** CAD, coronary artery disease; HF, heart failure; LVEDP, left ventricular end diastolic pressure; ML, machine learning; NRI, Net Reclassification Index; OVG, orthogonal voltage gradient; PPG, photoplethysmography.

paired with LVEDP (IDENTIFY; NCT #03864081). The primary objective was to validate a ML algorithm for prediction of elevated LVEDP using a definition of $\geq$25 mmHg (study cohort) and normal LVEDP $\leq$ 12 mmHg (control cohort), using AUC as the measure of diagnostic accuracy. Secondary objectives included performance of the ML predictor in a propensity matched cohort (age and gender) and performance for an elevated LVEDP across a spectrum of comparative LVEDP (<12 through 24 at 1 mmHg increments). Features were extracted from the OVG and PPG datasets and were analyzed using machine-learning approaches.

## Results

The study cohort consisted of 684 subjects stratified into three LVEDP categories, $\leq$12 mmHg (N = 258), LVEDP 13–24 mmHg (N = 347), and LVEDP $\geq$25 mmHg (N = 79). Testing of the ML predictor demonstrated an AUC of 0.81 (95% CI 0.76–0.86) for the prediction of an elevated LVEDP with a sensitivity of 82% and specificity of 68%, respectively. Among a propensity matched cohort (N = 79) the ML predictor demonstrated a similar result AUC 0.79 (95% CI: 0.72–0.8). Using a constant definition of elevated LVEDP and varying the lower threshold across LVEDP the ML predictor demonstrated and AUC ranging from 0.79–0.82.

## Conclusion

The phase space ML analysis provides a robust prediction for an elevated LVEDP at the point-of-care. These data suggest a potential role for an OVG and PPG derived electro-mechanical pulse wave strategy to determine if LVEDP is elevated in patients with symptoms suggestive of cardiac disease.

## Introduction

'Phase space' is a concept based on dynamical systems theory in which possible states of a given object such as position and velocity are represented with each state corresponding to one unique point in phase space [1]. While originating from mechanical systems, it has application to cardiovascular physiology. In one possible application, while systolic dysfunction is characterized by reduced ejection fraction, additional modalities are required to adjudicate dysfunction that is limited to diastole, with the aim of estimating left ventricular (LV) filling pressures. Left ventricular end diastolic pressure (LVEDP) is of distinct interest. The measurement of LVEDP, whether in the presence of reduced or preserved ejection fraction is complex and commonly characterized by multimodality diagnostic imaging. For example, elevation in Brain Naturetic Peptide (BNP) [2, 3] and fixed ratios based on echocardiography (spectral Doppler and Tissue Doppler derived E/e') [4] are used to classify if left atrial pressure is elevated or not. Several recent studies have aimed to predict diastolic dysfunction (i.e., intracardiac pressure elevation) using ML approaches, such as from CNN analysis of echocardiographic beat variability [5] and clustering of echocardiographic markers to understand the patterns of diastolic dysfunction across patients with symptomatic CVD [6]. While such developments are promising in the characterization of myocardial function, the prediction of LV

pressure elevation as a binary classification (elevated or not elevated) across a spectrum of LV pressures that can be used to guide downstream testing and treatment is of value.

In this context, phase space is a continuous measurement that simultaneously captures data related to electromechanical and pulse-wave signals over successive cardiac cycles, with the resultant biopotential plot being a large-scale data representation of myocardial function and is unique for any given person [1]. The benefits of such an approach are that it captures signals of myocardial function and dysfunction through high fidelity, time-series data collection that cannot be quantified by conventional non-invasive imaging or laboratory testing modalities [7, 8].

The physiologic findings of the failing heart that result in elevations in LV filling pressures, LV end diastolic pressure (LVEDP) and left atrial pressure are commonly determined by electrocardiographic and echocardiographic findings of atrial and ventricular remodeling [9], functional changes in diastolic relaxation [10], and changes in flow dynamics [5]. Given a highly heterogenous association between symptoms and the presence of cardiac dysfunction, especially in prevalent conditions such as those with heart failure with preserved EF (HFpEF), new modalities that leverage machine learning (ML) have emerged as potential tools to predict diastolic properties [11, 12] and ejection fraction (EF) [13] through computational approaches including neural network analysis of electrocardiographic intervals and wavelet transformation to predict myocardial function and relaxation.

Similarly, we have previously demonstrated the diagnostic accuracy of a ML approach to predict obstructive coronary artery disease (>70% luminal stenosis) from a cardiac phase space analysis. The predictive algorithm was trained and validated with tomographic, voltage-gradient features that were paired with the degree of coronary stenosis defined at angiography [7]. An approach such as this provided a method to collect data at the time of an outcome of interest and provided a pathway to evaluate cardiac dysfunction at the point-of-care [8, 14, 15]. In this context, we investigated a novel ML algorithm based on electromechanical features that was derived non-invasively from orthogonal voltage gradient (OVG) and photoplethysmography (PPG) to predict an elevated LVEDP among symptomatic patients referred for cardiac catheterization, and herein report the findings from the multi-center, prospective validation cohort.

## Results

### Study population

Overall, 1,023 study subjects were consecutively enrolled (Fig 1). After exclusions, the final study cohort consisted of 684 subjects with paired signals to LVEDP measurements and stratified into three LVEDP categories, ≤12 mmHg (N = 258), LVEDP 13–24 mmHg (N = 347), and LVEDP ≥25 mmHg (N = 79). Histogram of LVEDP in the study population can be found in Fig 2. Signals were acquired by trained nursing staff in a hospital-based setting (catheterization holding, cardiac unit, or observation area) in 671 (98.1%) or a physician's clinic in 13 (1.9%), within 7-days prior to angiography [630 (92%) on the same-day; 54 (8%) from 1–7 days].

Demographics characteristics for the overall population, study cohort as well as LVEDP groups are listed in Table 1. Within the study cohort, the mean age was 63 years and 45% were women. One third of the population had diabetes with greater than 70% with hypertension and/or hyperlipidemia. The mean EF was 60% with 93% (N = 186) with preserved EF >50%. 38% had obstructive CAD at angiography. Multivariate clinical predictors of an elevated LVEDP can be found in S1 File.

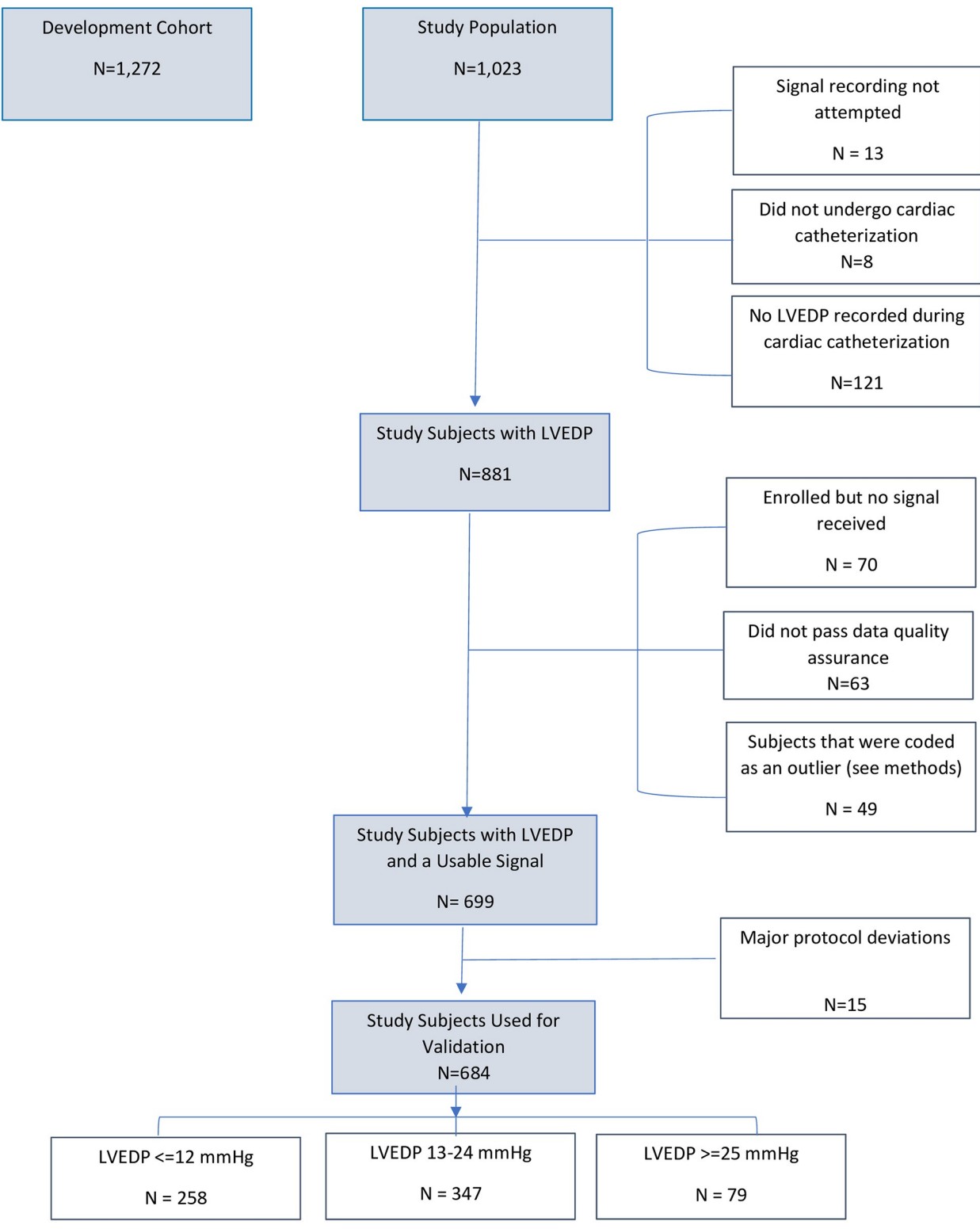

**Fig 1. Flow of subjects in the study.**

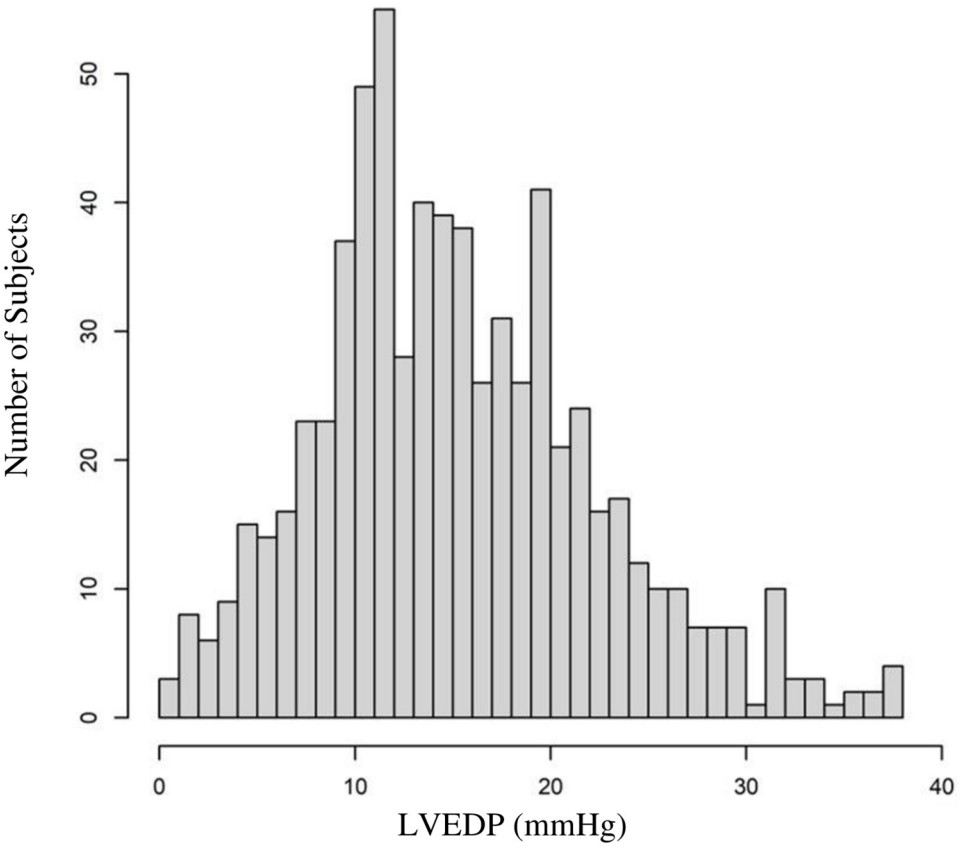

**Fig 2. Histogram of study population LVEDP.**

## Permutation feature importance and exemplar features

A permutation analysis was performed to determine feature importance in LVEDP elevation prediction, and the top 30 most contributive features were grouped by family (S2 File). The most contributive signal-based feature family was PPG indicators, followed by OVG spectral and phase space analysis. The most contributive feature within the PPG indicator family was the maximum of the PPG pulse base (S3 File). Fig 3 illustrates four exemplar features included in the machine-learned predictor, including the atrial depolarization duration, PPG pulse base, and ventricular repolarization. Detailed descriptions of the other features of importance can be found in S3 File.

## Primary outcome—Performance of the machine-learned predictor

All results (primary and secondary) used the ensembled model as a single assessment of algorithm performance on the blinded validation cohort. Testing of the machine-learned predictor as a continuous measurement demonstrated an AUC of 0.81 (95% CI 0.76–0.86) for algorithmic performance (Fig 4) and corresponded to a sensitivity and specificity of 82% (95% CI: 72–90%) and 68% (95% CI: 61–72%), respectively. S4 File contains the 2x2 cross tabulation for the prediction of an elevated LVEDP based on the sensitivity and specificity determined from the primary analysis.

**Table 1. Demographics.**

| Characteristic | Overall Study Population Development + Validation N = 1,956 | Development Cohort N = 1,272 | Validation Cohort N = 684 | p-value Development vs. Validation | Study Cohort: LVEDP≤12 or LVEDP≥25 N = 337 | LVEDP≤12 N = 258 | LVEDP≥25 N = 79 | LVEDP 13–24 N = 347 | p-value LVEDP ≤12 vs ≥25 |
|---|---|---|---|---|---|---|---|---|---|
| **Clinical Variables** | | | | | | | | | |
| Age (years) | | | | | | | | | |
| Mean ± STD | 52 ± 19 | 46 ± 20 | 63 ± 10 | <0.001 | 63 ± 11 | 64 ± 10 | 61 ± 12 | 62 ± 10 | 0.04 |
| ≥ 60 years (%) | 880 (45%) | 445 (35%) | 321 (47%) | <0.001 | 172 (51%) | 137 (53%) | 34 (43%) | 153 (44%) | 0.13 |
| Range | 18–91 | 18–88 | 30–91 | | 30–90 | 36–90 | 30–83 | 31–91 | |
| Ejection Fraction | | | | | | | | | |
| Measured (%) | 978 (50%) | 560 (44%) | 428 (63%) | - - | 199 (59%) | 157 (61%) | 43 (54%) | 229 (66%) | 0.60 |
| Mean ± STD | 59 ± 8 | 59 ± 8 | 60 ± 7 | - - | 60 ± 7 | 60 ± 6 | 59 ± 10 | 60 ± 8 | 0.56 |
| >50% | 892 (91%) | 501 (90%) | 391 (91%) | - - | 186 (93%) | 148 (94%) | 38 (90%) | 205 (90%) | 0.81 |
| ≤ 50% | 90 (9%) | 53 (10%) | 37 (9%) | - - | 13 (7%) | 9 (6%) | 4 (10%) | 24 (10%) | 0.81 |
| ≤ 40% | 40 (4%) | 27 (5%) | 13 (3%) | - - | 6 (3%) | 3 (2%) | 3 (7%) | 7 (3%) | 0.31 |
| ≤ 30% | 19 (2%) | 13 (2%) | 6 (1%) | - - | 3 (2%) | 1 (1%) | 2 (5%) | 3 (1%) | 0.29 |
| Women (%) | 911 (47%) | 612 (48%) | 299 (44%) | 0.070 | 152 (45%) | 101 (39%) | 51 (65%) | 147 (42%) | <0.001 |
| History of Hypertension (%) | 1014 (52%) | 519 (41%) | 495 (72%) | <0.001 | 244 (72%) | 184 (71%) | 60 (76%) | 251 (72%) | 0.51 |
| History of Hyperlipidemia (%) | 1005 (51%) | 512 (40%) | 493 (72%) | <0.001 | 246 (73%) | 194 (75%) | 52 (66%) | 247 (71%) | 0.12 |
| History of Diabetes (%) | 460 (24%) | 246 (19%) | 214 (31%) | <0.001 | 102 (30%) | 75 (29%) | 27 (34%) | 112 (32%) | 0.48 |
| Smoker* (%) | 728 (37%) | 376 (30%) | 352 (51%) | <0.001 | 178 (53%) | 139 (54%) | 39 (49%) | 174 (50%) | 0.54 |
| Body Mass Index | | | | | | | | | |
| Mean ± STD | 30 ± 7 | 29 ± 7 | 32 ± 7 | <0.001 | 32 ± 7 | 30 ± 6 | 37 ± 8 | 32 ± 6 | <0.001 |
| ≥ 30 (%) | 899 (46%) | 520 (41%) | 379 (55%) | <0.001 | 176 (52%) | 111 (43%) | 65 (82%) | 203 (59%) | <0.001 |
| <30 | 1056 (54%) | 751 (59%) | 305 (45%) | <0.001 | 161 (48%) | 147 (57%) | 14 (18%) | 144 (41%) | <0.001 |
| **Race/Ethnicity** | | | | | | | | | |
| Caucasian (%) | 1440 (74%) | 861 (68%) | 579 (85%) | <0.001 | 280 (83%) | 221 (86%) | 59 (75%) | 299 (86%) | 0 |
| Black or African American (%) | 396 (20%) | 315 (25%) | 81 (12%) | <0.001 | 44 (13%) | 27 (10%) | 17 (22%) | 37 (11%) | 0 |
| Other Races (%) | 120 (6%) | 96 (8%) | 24 (4%) | <0.001 | 13 (4%) | 10 (4%) | 3 (4%) | 11 (3%) | 0 |
| **Symptoms** | | | | | | | | | |
| Chest Pain During Exercise | 577 (29%) | 384 (30%) | 384 (56%) | <0.001 | 193 (57%) | 147 (57%) | 46 (58%) | 191 (55%) | 0.95 |
| Chest Pain at Rest | 404 (21%) | 266 (21%) | 282 (41%) | <0.001 | 138 (41%) | 99 (38%) | 39 (49%) | 144 (41%) | 0.11 |
| Dyspnea During Exercise | 646 (33%) | 438 (34%) | 426 (62%) | <0.001 | 208 (62%) | 153 (59%) | 55 (70%) | 218 (63%) | 0.13 |
| Dyspnea at Rest | 199 (10%) | 127 (10%) | 145 (21%) | <0.001 | 72 (21%) | 49 (19%) | 23 (29%) | 73 (21%) | 0.08 |
| **Medications** | | | | | | | | | |
| ACE inhibitor[†] | 378 (19%) | 185 (15%) | 193 (28%) | <0.001 | 92 (27%) | 72 (28%) | 20 (25%) | 101 (29%) | 0.76 |
| ARB[†] | 302 (15%) | 148 (12%) | 154 (23%) | <0.001 | 77 (23%) | 55 (21%) | 22 (28%) | 77 (22%) | 0.29 |
| Diuretic[††] | 323 (17%) | 187 (15%) | 136 (20%) | 0.004 | 65 (19%) | 36 (14%) | 29 (37%) | 71 (20%) | <0.001 |
| Calcium Channel Blocker[†] | 313 (16%) | 163 (13%) | 150 (22%) | <0.001 | 70 (21%) | 48 (19%) | 22 (28%) | 80 (23%) | 0.11 |
| Beta Blocker | 563 (29%) | 286 (22%) | 277 (40%) | <0.001 | 142 (42%) | 102 (40%) | 40 (51%) | 135 (39%) | 0.11 |

*(Continued)*

**Table 1.** (Continued)

| Characteristic | Overall Study Population Development + Validation N = 1,956 | Development Cohort N = 1,272 | Validation Cohort N = 684 | p-value Development vs. Validation | Study Cohort: LVEDP≤12 or LVEDP≥25 N = 337 | LVEDP≤12 N = 258 | LVEDP≥25 N = 79 | LVEDP 13–24 N = 347 | p-value LVEDP ≤12 vs ≥25 |
|---|---|---|---|---|---|---|---|---|---|
| Aldosterone Receptor Antagonist | 9 (0%) | 0 (0%) | 9 (1%) | <0.001 | 4 (1%) | 3 (1%) | 1 (1%) | 5 (1%) | 1.00 |
| Statins | 791 (40%) | 388 (31%) | 403 (59%) | <0.001 | 200 (59%) | 158 (61%) | 42 (53%) | 203 (59%) | 0.25 |
| **Angiographic Findings** | | | | | | | | | |
| Presence of obstructive CAD^ | 497 (36%) | 253 (36%) | 244 (36%) | - - | 129 (38%) | 110 (43%) | 19 (24%) | 115 (33%) | 0.005 |
| **Hemodynamic Variables** | | | | | | | | | |
| LVEDP | | | | | | | | | |
| Mean ± STD | 16 ± 7 (N = 1380) | 16 ± 7 (N = 696) | 16 ± 7 | - - | 14 ± 9 | 9 ± 3 | 29 ± 4 | 18 ± 3 | <0.001 |
| Range | 0–45 | 0–45 | 0–38 | | 0–38 | 0–12 | 25–38 | 13–24 | |
| LVSP | | | | | | | | | |
| Mean ± STD | 129 ± 24 (N = 1002) | 130 ± 24 (N = 395) | 129 ± 24 (N = 607) | - - | 126 ± 25 (N = 283) | 119 ± 21 (N = 208) | 143 ± 25 (N = 75) | 132 ± 23 (N = 324) | <0.001 |
| Range | 15–221 | 40–221 | 15–209 | | 22–209 | 22–209 | 88–196 | 15–208 | |

Ejection fraction measured and reported by ventriculography at the time of angiography

*past or present smoker

†including combination medications

††Avalide, Dyazide, Furosemide, Hydrochlorothiazide, Maxzide, Triamterene

^at least one coronary lesion with a stenosis of ≥70% and/or FFR<0.8 and/or iFR <0.89 on angiography

## Secondary outcomes

**Predictive performance for an elevated LVEDP among a propensity matched cohort.** Among a propensity-matched cohort (N = 79 pairs of study subjects) between elevated and non-elevated LVEDP based on age and gender (S5 File), the machine-learned predictor demonstrated a similar result to the primary analysis for the prediction of an elevated LVEDP, AUC 0.79 (95% CI: 0.72–0.86, Fig 5).

**Determine the predictive performance for an elevated LVEDP across a spectrum of comparative LVEDP thresholds.** Using a constant elevated definition of LVEDP ≥ 25 mmHg and varying the definition of non-elevated across LVEDP values (in 1 mmHg increments), the machine-learned predictor demonstrated an AUC ranging from 0.79–0.82 (Fig 6a) with corresponding specificity between 59%-69% (using the constant, predefined elevated LVEDP threshold yielding a sensitivity of 82%). Fig 6b illustrates the effect of varying both the definitions of LVEDP elevation and non-elevated, demonstrating a consistent performance, based on AUC, for the prediction of an elevated LVEDP at a threshold value of 25 mmHg.

**Sub-group performance.** Among the predefined sub-groups, the machine-learned predictor demonstrated an adequate diagnostic accuracy between group stratifications (Fig 7). While the predictive accuracy for an elevated LVEDP was similar among cohorts with obstructive and non-obstructive (p = 0.31), there was a statistically significant difference with greater predictive accuracy among cohorts with preserved EF compared to low EF (p = 0.03).

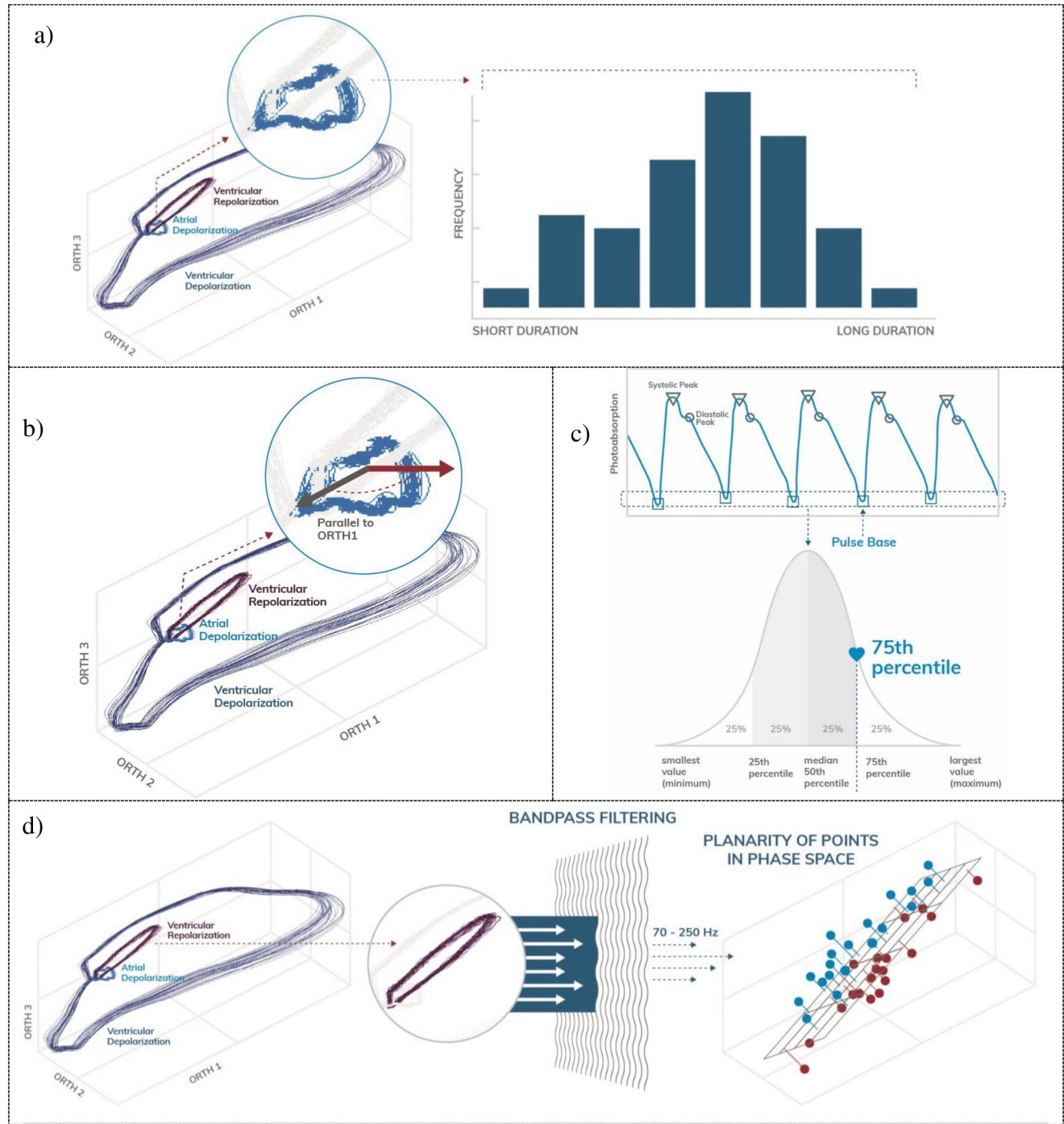

**Fig 3. Exemplar features.** a) the variation in the atrial depolarization duration is a feature extracted from the OVG signal in the time domain, b) quantification of the atrial depolarization vector in phase space, c) quantification of PPG pulse base amplitude and d) ventricular repolarization in band-pass filtered phase space.

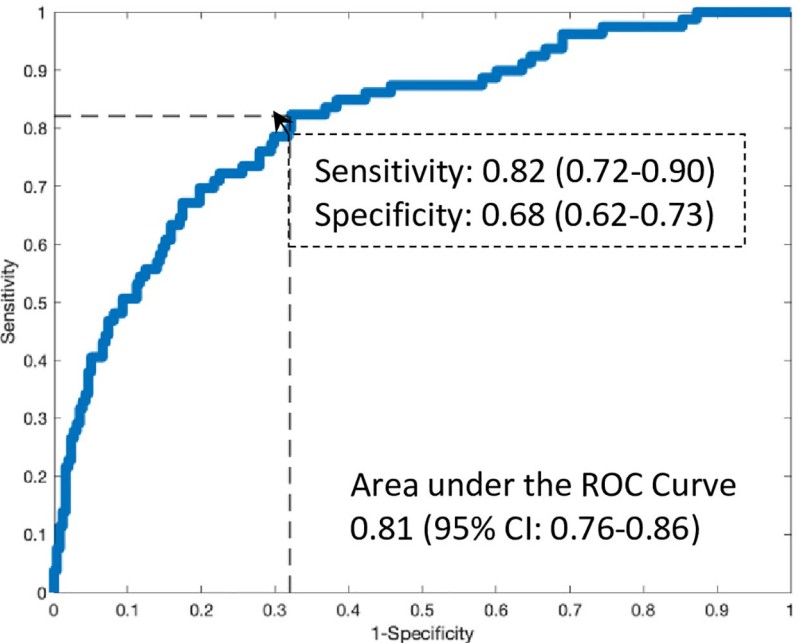

**Fig 4. Performance of the machine-learned predictor.**

## Safety and adverse events

Testing of the machine-learned model within the pre-specified safety analysis among a healthy cohort without CV disease, at a threshold sensitivity of 82% reported in the primary analysis, demonstrated a specificity of 95% (95% CI: 90–97%). The corresponding 2x2 tabulation and AUC can be found in S4 File.

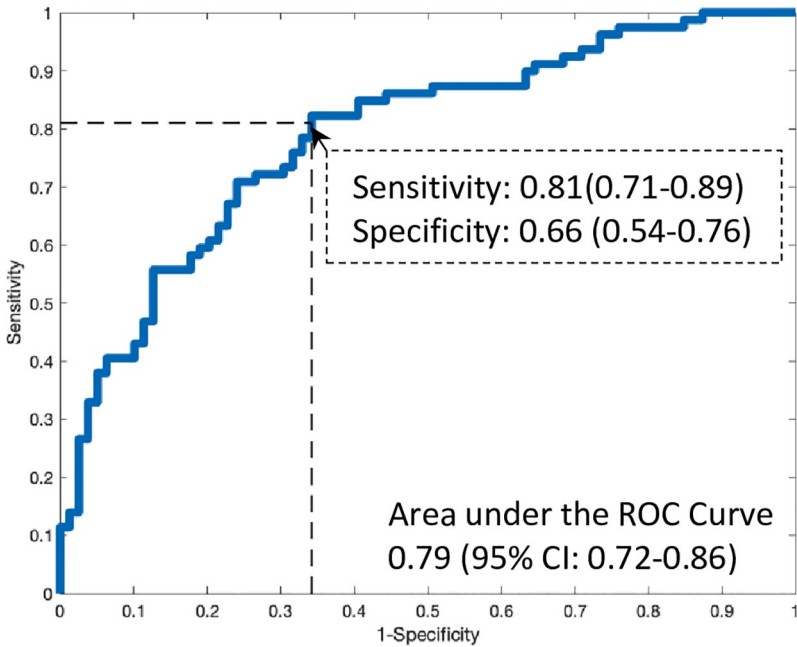

**Fig 5. Performance of the machine-learned model when propensity matching based on age and gender.**

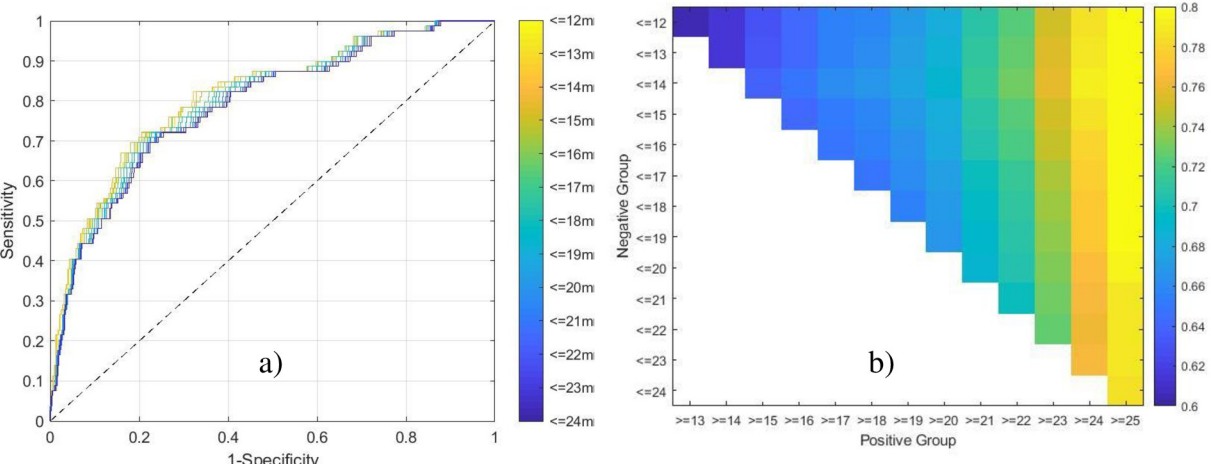

**Fig 6. Performance while varying the definition of LVEDP elevation and non-elevation.** a) the definition of elevation is held constant at ≥25mmHg, while the definition of non-elevation is varied from ≤12mmHg to ≤24mmHg, and the ROC curves assessed for each scenario. b) the definition of both elevation and non-elevation are varied from ≥13-25mmHg and ≤12-24mmHg, respectively, and the corresponding AUCs represented as a heatmap.

No adverse events related to device use were reported during the study.

## Bayesian analysis and simulations for net reclassification index between BNP and the machine-learned predictor

A Bayesian analysis of the post-test probability of an elevated LVEDP based on a range of pre-test probabilities and distributed according to the machine learned predictor, and the two

| Subgroup | AUC (95% C.I.) | N | p-value |
|---|---|---|---|
| Age < 60 | 0.85 (0.76, 0.93) | 119 | 0.2786 |
| Age >= 60 | 0.79 (0.71, 0.86) | 218 | |
| BMI < 30 | 0.71 (0.56, 0.86) | 161 | 0.6412 |
| BMI >= 30 | 0.75 (0.67, 0.83) | 176 | |
| Diabetic | 0.80 (0.71, 0.90) | 102 | 0.9191 |
| Non-diabetic | 0.81 (0.74, 0.88) | 234 | |
| Women | 0.80 (0.73, 0.88) | 152 | 0.7285 |
| Men | 0.78 (0.69, 0.88) | 185 | |
| Non-obstructive CAD on angiography | 0.83 (0.76, 0.89) | 208 | 0.3060 |
| Obstructive CAD on angiography | 0.76 (0.65, 0.87) | 129 | |
| Ejection Fraction <= 50% | 0.44 (0.11, 0.77) | 13 | 0.0349 |
| Ejection Fraction > 50% | 0.85 (0.78, 0.92) | 186 | |
| Caucasian | 0.80 (0.74, 0.87) | 280 | 0.9776 |
| Black or African American | 0.80 (0.67, 0.94) | 44 | |
| Sites enrolling less than 50 subjects | 0.80 (0.73, 0.87) | 230 | 0.5496 |
| Sites enrolling more than 50 subjects | 0.83 (0.73, 0.93) | 107 | |

**Fig 7. Subgroup analyses of the performance of the machine-learned predictor.**

simulated BNP performances in a matching population–a BNP threshold of 150 pg/ml and a BNP threshold of 50 pg/ml—are illustrated in Fig 8a–8c; respectively. Using the approximate values of the constraining statistics for the two BNP data sets (non-cardiac symptoms and obese HFpEF, see Methods), the simulation yielded an AUC of 0.69 (95% CI: 0.61–0.71) for BNP prediction of an elevated LVEDP with the corresponding sensitivity, specificity, positive and negative likelihood ratios for the thresholds of a BNP of 150 and 50 pg/ml reported within Fig 8. The NRI of the ML predictor using a posterior probability of a BNP at a threshold of 150 and 50 pg/ml was 0.24 (95% CI: 0.18–0.30) and 0.38 (95% CI: 0.33–0.44), respectively.

## Discussion

There is a growing need for new methods to measure LV filling pressures. Recent studies have used ML approaches that analyze echocardiography data to predict diastolic dysfunction [5, 6]. While the prospect for discovery is promising, the application of any new analytic technique requires robust methodologies for validation. In this context we employed a trial design of prospective validation [16] within a multicenter study. Prospective data collection permitted the validation dataset to be blinded from the training dataset. This is important because blinding may limit common biases such as spectrum bias and measurement bias between training and validation datasets, and to balance clinical characteristics between both datasets [14]. Towards mitigating bias, we recruited a diverse cohort of patients across multiple healthcare centers and geographies and aimed to enroll study subjects that are representative of a real-world population with a clustering of cardiac risk factors, and various ethnicities. Overall, half of the participants recruited were women, nearly 50% with a BMI ≥ 30 (mean of 36) and greater than 90% with preserved EF (mean EF of 61%), a triad of findings where an accurate assessment of LV filling pressures by conventional testing such as BNP [17] and echocardiography [4, 18] have marginal accuracy and vary significantly across those with symptoms of HF. This is particularly true in HFpEF given the heterogeneity of myocardial dysfunction (i.e. ischemic vs non-ischemic etiologies), the cardiopulmonary response to increased afterload and/or preload [19] and the phasic changes in left atrial function that are variable across individuals [9]. Our observations of a high incidence (34%) of an elevated LVEDP among symptomatic patients referred to angiography for the evaluation of ischemic heart disease; however, did not have evidence of obstructive CAD is equally important as it may reflect the underdiagnosis of HF in an ambulatory cohort, and those referred for further cardiovascular testing.

Our hypothesis that electromechanical pulse wave features predict myocardial dysfunction is an extension of the hypothesis that the progression from normal myocardial mechanics to pathologic atrial and ventricular remodeling is a result of rising LV filling pressure. Whereas atrial enlargement and ventricular remodeling from alternations in myocardial tension and strain can be considered mechanical features of a pressure loaded left ventricle [20], we postulate that a high dimensional dataset captured from voltage gradients and photoplethysmography can accurately represent myocardial electromechanical function. In support of this argument, wavelet transformation and the mathematical conversion of an ECG into a normalized energy distribution (depicted by a color spectrum of myocardial energy) has recently emerged as a computational modeling and ML method for the prediction of diastolic dysfunction. Potter [11] and Sengupta [12], paired ECG data with echocardiographic data of diastolic abnormalities such as E/e'> 14, left atrial enlargement, and abnormal LA volume index among 398 and 188 patients at risk of HF, respectively. Using supervised and unsupervised ML including random forest classifiers on 250–650 wavelet features, testing of the ML algorithm on a validation data set demonstrated a high diagnostic accuracy (AUC 0.83–0.91) for the

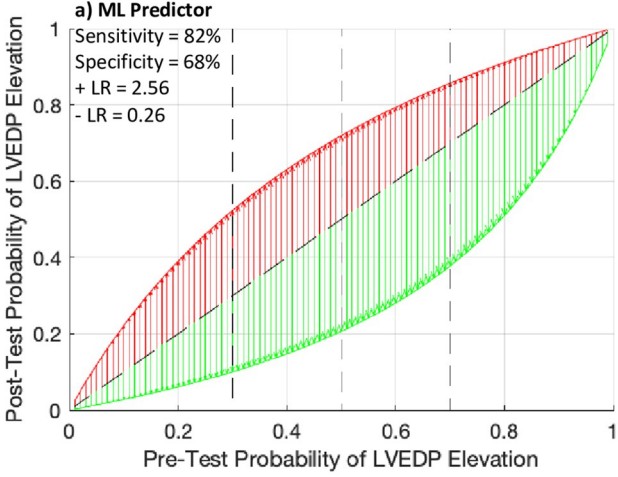

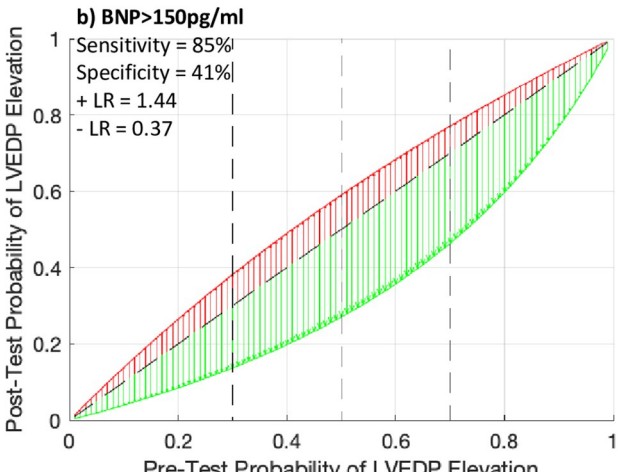

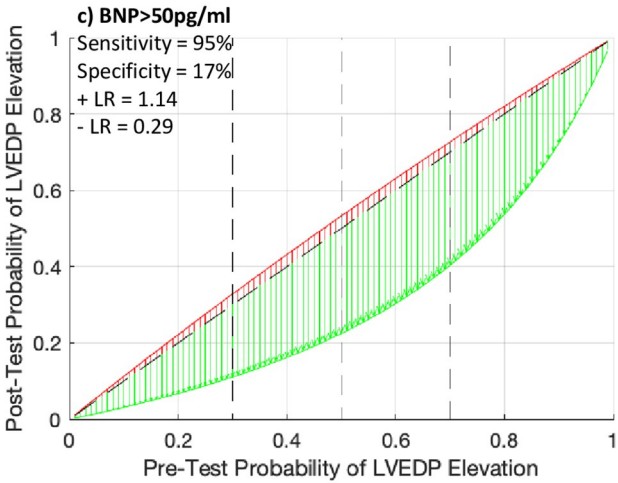

**Fig 8. Relationship between pre-test (prior) probability and post-test (posterior) probabilities.** a) the machine-learned predictor, b) BNP when greater than 150pg/ml, or c) BNP when greater than 50pg/ml. A positive test is shown in red, and a negative test in green. The diagonal dashed black line represents no change from the pre-test to post-test probability. The vertical dashed black lines represent intermediate to high pre-test probabilities (vertical dashed lines at 30%, 50% and 70%) from left to right. The post-test probabilities were calculated based on a varying pre-test probability, and constant sensitivity, specificity, and corresponding likelihood ratios.

prediction of diastolic dysfunction, with a diagnostic performance that was greater than clinical prediction alone.

The present study extends Potter and Sengupta's results to a time-series, electromechanical and perfusion dataset with training and validation on direct LVEDP measurements. Within our dataset, in an LV with normal EF that is under high pressure [10], electromechanical features such as variation in atrial depolarization duration, ventricular repolarization in phase space plausibly represents physiologic findings of elevated left atrial pressure [9] and lusitropic changes of diastolic relaxation [11, 12]; respectively. PPG derived feature and the measurement of the pulse wave base may represent a systolic time interval of isovolumetric contraction as this nadir point in the pulse-wave is associated with the lowest photoabsorption immediately prior to onset of systole; a time point of interest that has been associated with HF [21]. Our method of high frequency data capture analyzed in phase space and the corresponding electro-mechanical features are unique in any given study subject. Once all features are evaluated from a patient's signal, the machine-learned model processes the feature values to yield a continuous score representing the risk of LVEDP elevation. In contrast to BNP or echocardiography which use a threshold or binary values of elevated or non-elevated LV pressures, the high number of features used in this analysis creates a unique signature of LVEDP that is specific to that individual at the N-of-1 level.

One pertinent question leading from the present analysis, is how our results are translated within the continuum of diagnostic tests to determine the presence of elevated LV pressure among symptomatic patients, particularly those with preserved EF. Several point-of-care diagnostic tests are available to diagnose acute HF including chest radiography, BNP/NT-pro-BNP, handheld echocardiography/lung ultrasound, and bioimpedance [22]. Various studies evaluating such point-of-care tests have largely concluded that lung ultrasound and echocardiography have utility to differentiate HF symptoms from non-HF symptoms [22]. While useful, these tests require trained individuals to acquire and interpret cardiopulmonary images and can have limited diagnostic accuracy as they are dependent on patient characteristics such as body habitus and the user experience with, point-of-care imaging devices. Our findings of high diagnostic accuracy for prediction of elevated LVEDP when compared to a range of non-elevated LVEDP such as normal ($\leq$12 mmHg) and mid-range (13–24 mmHg) is potentially valuable for 2 main reasons: 1) it supports that our method for data capture and analysis of electromechanical data is robust and that our features are those data representations associated with elevated LVEDP; and 2) that LVEDP at a threshold of $\geq$25 mmHg is representative of an elevation that is clinically relevant as our study population was derived from symptomatic patients requiring cardiac catheterization. The latter is important in the setting of high-risk HF patients such as those at risk for HF re-hospitalizations. If we assume that such patients have, at minimum, a 50% (intermediate) to 70% (high) pre-test probability of an elevated LVEDP, when used sequentially with BNP, our Bayesian simulation demonstrate that the ML predictor reclassifies a BNP of > 150 pg/ml (NRI of 0.24) from 59–77% to a post-test probability of 79–90%, with the greatest reclassification margin within the intermediate (30–50%) pre-test probability group. Such results may have clinical utility to more accurately triage HF patients at the point-of-care for further testing and to identify patients with HFpEF.

## Limitations

We identified 14% (36/258) of subjects with non-elevated LVEDP were taking a diuretic at the time of enrollment that may have impacted the performance of the ML predictor. When compared to overall study population the specificity was similar within this cohort (68% vs 64%, p = 0.57) and when the analysis was re-run when excluding this cohort, there was no difference

in overall specificity. While we contend that diuretics are an important factor when considering the measurement and prediction of LVEDP, the small number of subjects in this group does not permit us to determine its impact on performance within the study population as presented.

Our study population is intrinsically limited by the recruitment methodology, which was subjects referred to left heart catheterization for assessment of obstructive CAD using coronary angiography, and specifically the subgroup where the treating physician chose to measure the LVEDP. We employed this study methodology to ensure that subjects had a catheterization-confirmed elevated LVEDP, but at the limitation of subjects referred for the evaluation of obstructive CAD. While this may introduce sample bias, we found significant CAD in only 38% of the overall study cohort, a higher incidence of obstructive CAD was observed in subjects with non-elevated LVEDP compared to those with elevated LVEDP (43% vs 24%). Upon subgroup analysis, there was difference in algorithmic performance among those with or without obstructive CAD.

Overfitting, and conversely generalizability, are critical aspects of machine learning and when a large number of features are used for model development. The use of an ensemble, as is the case, does not increase the likelihood of overfitting but rather mitigates it by reducing the dependence on a single model. The methods employed to avoid overfitting could include the use of cross-validation within the development data, using simple models with regularization and penalty terms, and testing the performance of the model on unseen data (doing so only once). With respect to the models in particular, first, each model is exposed to an average of 149 features (with a range of 89–194, S10 File). Second, the model hyperparameters were conservatively designed to mitigate the possibility of overfitting (S10 File). For example, four of the models were Random Forest, which intrinsically limit overfitting by only allowing each component tree access to the square root of the total number of features, and by bootstrap sampling training subjects so that every component tree only has access to a subset of the entire training set. Overfitting was additionally controlled through the use of the maximum tree depth hyperparameter. Deep trees with many splits increase the likelihood of overfitting, and therefore the depth was limited to 3–7. Other model types (Elastic Net and XGBoost) were also designed conservatively. Finally, the ultimate test of overfitting is the performance on unseen blinded data, which yielded an AUC of 0.81. In conclusion, through the analysis of the algorithm, and the performance on unseen blinded data, overfitting did not occur.

## Conclusions

We validated a machine learning algorithm of electromechanical pulse wave features to predict an elevated LVEDP among symptomatic patients with a precise measurement of LVEDP. Such techniques to quantify intracardiac pressure with machine learning on large datasets acquired with a portable digital device provides a new method to determine the presence or absence of HF. These data suggest a potential role for a novel OVG and PPG derived electro-mechanical diagnostic test for the prediction of an elevated LVEDP at the point-of-care.

## Methods

### Trial design

Enrollment in the overall IDENTIFY trial began on December 10[th] 2018 with 3,486 participants consecutively enrolled as of April 2021 and was primarily executed to develop a machine-learned predictor to determine the presence of obstructive coronary artery disease (CAD) defined at cardiac catheterization. A cohort analysis (N = 606) using a phase-space ML approach to predict CAD using the same inclusion/exclusion criteria as in this present study

has been previous published [7]. The present results are reported according to STARD guidelines [23] (S6 File). The study was approved by a centralized IRB (Western IRB #20183107, now known as WIRB-Copernicus Group). It was initially released on clinicaltrials.gov on January 1st 2019 (NCT #03864081), and was performed at 15 healthcare institutions in the United States (S7 File). A preliminary analysis of the LVEDP development cohort was presented at the 2020 Scientific Sessions of the American College of Cardiology [24].

## Study population—Development and validation data

**Data sources.** The data sources for the study population included patients who were referred for angiography at the discretion of their treating physicians and for the evaluation of symptoms suggestive of CAD. Patients provided written informed consent to participate in the study. Inclusion and exclusion criteria have been previously published [7] and can be found in S8 File.

**Development and validation groups.** The study population was derived from a pooled individual patient-level analysis stratified by unique time points and into development and validation cohorts. The development cohort was comprised of symptomatic patients referred to cardiac catheterization for the evaluation of CAD and consecutively enrolled between April 2017 –December 2017 (N = 696) and included asymptomatic individuals without CVD (N = 576). A separate validation cohort prospectively consecutively enrolled symptomatic patients referred to cardiac catheterization between March 2019 –November 2019 (N = 1,023). The data sources for the development and validation cohorts were separate with no data from the validation group used for development. Therefore, the validation group is considered blinded.

## Primary and secondary study objectives

The primary objective was to develop and validate an ML algorithm for the prediction of an elevated LVEDP $\geq$ 25 mmHg. LVEDP was measured invasively at the time of cardiac catheterization using conventional techniques for left ventricular pressure assessments. For the primary objective, an analysis using threshold of LVEDP of 25 mmHg (study cohort) was chosen and compared to individuals with a normal LVEDP defined as $\leq$ 12 mmHg (control cohort). These thresholds were selected to reflect those LVEDP measurements that are likely to be sufficiently high to result in symptoms (elevated LVEDP) or absence of symptoms (normal LVEDP), as they relate to a spectrum of symptoms among CV patients undergoing angiography, and those with obstructive and non-obstructive CAD [25, 26].

Secondary objectives included those analyses to refine the primary objective within the following 5 categories:

1. Performance of the machine-learned predictor for an elevated LVEDP $\geq$ 25 mmHg among a propensity-matched cohort (scoring using age and gender).

2. Performance of the machine-learned predictor for an elevated LVEDP across a spectrum of comparative LVEDP thresholds (<12 through 24 at 1 mmHg increments).

3. Sub-group performance stratified by age (<60 vs. $\geq$60), gender, comorbidities, EF (<50% vs $\geq$50%), CAD status determined at angiography [obstructive CAD (defined as $\geq$1 lesion with a stenosis of 70%; or $\geq$1 lesion with a fractional flow reserve of $\leq$0.80; or $\geq$1 lesion with an instantaneous wave-free ratio of $\leq$0.89) vs non-obstructive CAD], ethnicity, and enrollment site (stratified by number of study subjects enrolled N$\geq$50 vs N<50).

4. Safety analysis and predictive accuracy of the machine-learned predictor using a healthy control cohort to determine the specificity and negative predictive value of the algorithm, and;

5. Bayesian analysis to determine the post-test (i.e., posterior) probability of the machine-learned predictor based on varying the pre-test probability (i.e., low, intermediate, and high prior probability of elevated LV filling pressures) among symptomatic patients.

## Acquisition system description

The acquisition system (CorVista Capture™ device) simultaneously collects two modalities of time series data: orthogonal voltage gradient (OVG) data, representing cardiac electro-mechanical activity analyzed in phase space, and photoplethysmography (PPG) data representing blood volume changes as a measurement of distal perfusion.

Consecutive study subjects underwent signal acquisition immediately prior to angiography or within seven days prior to the procedure. In addition to OVG and PPG data, the device also captured patient-specific metadata (gender, age, height and weight). Signal data was acquired for 3.5 minutes.

## Raw data collection

The OVG signal is collected using electrodes attached to the skin (S9 File). Specifically, the signal is acquired at 8kHz (i.e., 8,000 samples per second, with each consecutive pair of samples separated by 0.000125 seconds) from seven electrodes at an amplitude resolution of 0.024 microvolts. The signal originates from three bipolar pairs of electrodes collecting data from the coronal, sagittal and transverse planes, and the seventh electrode acting as the reference. See S9 File for further details. Similar to existing signal collection methods, the OVG measures the biopotential at the surface of the skin caused by cardiac electrical activity. While the OVG signal acquisition resembles ECG, it differs with greater sampling frequency (conventional ECG sampling frequency of 500-2000Hz) by a factor of 4–16 and provides broader spatial information due to the orthogonal lead configuration and vectors along different planes of the body [27, 28].

The OVG biopotential data is represented within a three-dimensional phase space, where the parameters of the phase space are defined by the three bipolar orthogonal acquisition channels. Specifically, the amplitudes of three voltage gradient data points from the three channels form a three-dimensional coordinate within the phase space. As the signal processes through time, it traces a phase space trajectory. The PPG signal contains red and infrared light components, both collected at 500Hz via a finger clip sensor. The pulse wave is captured as the absorption of light in the tissue varies based on changes in cardiac activity.

## Development & validation approach for the machine learning predictor

Development and validation of a machine learning predictor occurred in two distinct phases. In the first phase, the machine learning predictor was trained using the development dataset. Upon completion, the machine-learned predictor was finalized such that no further modifications were permitted. Then, in the second phase, the machine learning predictor was tested in the blinded validation cohort and the performance was assessed.

## Signal processing and development of the machine-learned predictor

The sequence for processing a patient's data to generate the ML prediction occurred in four steps.

**Step 1: Confirmation of signal quality.** As an initial processing stage, the signal was confirmed to have adequate quality to proceed through the next three steps. Signal quality acts to check for the presence of noise generated by common sources in a clinical environment. The signal quality assessment has been previously published [8], and will be summarized herein. The OVG signal is examined for the presence of powerline noise, which is the electrical noise at the frequency at which alternating current (AC) power is delivered (i.e., to electrical outlets, etc.); specifically, this is 60 cycles per second (Hz) in North America. The OVG signal is also examined for high-frequency noise. A SNR of 57 was considered acceptable for powerline noise, and of 19 for high frequency noise. The PPG signal is examined for sensor saturation, which can occur when the light emitted from the LED on one side of the finger clip directly enters the sensor on the opposite site of the clip without transiting the finger. The light is strong because it isn't attenuated by the finger, and therefore an optical value is registered that exceeds the maximum measurable value. Excessive occurrence of this situation reduces the physiological information in the signal, and results in a prompt to attempt to reacquire the signal. SNR is not applicable to this score because the occurrence is transient.

**Step 2: OVG and PPG feature extraction.** After signal quality assessment, features were extracted from the signal. We defined a feature as a characteristic of the data that is automatically measurable on any acquired signal. A spectrum of feature domains of signal characterization were used in the present analysis and include the dynamics of the OVG and PPG signals in isolation; the synchronization dynamics of the OVG and PPG signals, the spectral properties of each signal modality; deviations of the OVG signal from subject-specific models; both conventional time-domain features and variations of those features; phase space features; PPG pulse-wave indicators; and approximation of a patient's respiration waveform (see S3 File). For example, the variation in the atrial depolarization duration is a feature extracted from the OVG signal in the time domain. Specifically, the duration of the atrial depolarization is measured on each cardiac cycle which forms a distribution of durations across the length of data acquisition. The standard deviation of this distribution is then calculated to represent the variation in the atrial depolarization duration. Examples of other features can be found following in S3 File.

The OVG data represents the entirety of the electrical biopotential signals plotted on axes corresponding to the signal amplitude in millivolts of each channel, where ventricular depolarization and repolarization, and atrial depolarization, appear visually as loops. While this OVG data may appear similar across different people, it is unique for a given individual, therefore generating unique feature vectors in the high-dimensional feature space.

**Step 3: Outlier detection.** Mathematically outlying subjects were identified based on the signal's feature values using the Isolation Forest algorithm [29]. Excluding outlying data ensures that the algorithm is not exposed to any data that is significantly differently than the development data.

**Step 4: Machine learned model optimization.** 13 machine-learned models optimized to return high values for elevated LVEDP and low values for non-elevated LVEDP were evaluated given the features as input. Each model was trained individually by varying the subjects, features and thresholds. The models also varied, from Random Forest [30], Extreme Gradient Boosting [31] and Elastic Net [32]. See S10 File for a description of the algorithm, data, and hyperparameters for each model, as well as an explanatory figure for Extreme Gradient Boosting. Each of the 13 models were individually performant based on stratified 5-fold cross-validation repeated for 100 iterations (to vary the train/test folds) within the development data (S11 File), but represent unique analyses of LVEDP assessment. To capture the diversity of each model in a final single prediction, which is intended to eliminate the bias associated with the selection of a single model and thus reduce the likelihood over overfitting on the

development data, the 13 model were amalgamated into a single predictive ensemble. The ensemble, composed of an average of the normalized outputs from the constituent models, is intended to on-average outperform any model that we may have selected from the pool of 13 when applied to new data [33].

## Statistical analysis

Analyses were performed to determine the diagnostic performance of the machine learned predictor used as a continuous measurement (independent variable) on the prediction of LVEDP (dependent variable) adjudicated as elevated (≥25mmHg) or normal (≤12mmHg). A threshold was then established on the continuous measurement to yield a binary output, after which standard techniques were used to calculate sensitivity, specificity, negative and positive predictive values with each study subject categorized as either a true negative, true positive, false negative, and false positive. R 3.5.2 was used for statistical calculations, including 95% CIs, relevant statistical tests, ROC-AUCs, and propensity matching between the elevated LVEDP and control cohort. CIs were calculated using De Long's method for AUC, and Clopper Pearson for sensitivity and specificity. Comparisons between ROC-AUCs were computed with DeLong's test. Independent clinical predictors for an elevated LVEDP were calculated using multivariable logistic regression analyses. Permutation analysis was used to determine feature contribution within the machine learning model [30].

A simulation of BNP performance was performed using the known distributions of measured BNP values [2, 3], and specifically the estimated minimums, 25th percentiles, medians, 75th percentiles and maximums [2, 3] Specifically, the performance of BNP, as a commonly used point-of-care test for the prediction of heart failure, was explored using published studies that included patients with similar clinical characteristics to those in the present study (symptoms suggestive of decompensated HF [3], those with HFpEF, and those with obesity [2]) and to compare the post-test probability of the machine-learned predictor vs BNP. This simulation was then used to explore the sequential prediction of an elevated LVEDP [2] and non-elevated LVEDP [3] (i.e. dyspnea due to non-cardiac causes) using the BNP post-test (i.e., posterior) probability as the pre-test (i.e., prior) probability for the machine-learned predictor. Therefore, to propose how the present results may be used in clinical practice, the sequence of this analysis is the following: pre-test probability → BNP post-test probability used as the machine-learned posterior probability → machine-learned predictor post-test probability → prediction of elevated LVEDP; and to determine a net reclassification index (NRI) within this sequence [34].

The simulation of BNP performance was performed using the distributions of measured BNP values in two relevant publications cohorts [2, 3], specifically the estimated minimums, 25th percentiles, medians, 75th percentiles and maximums using boxplots. The approximate values of the constraining statistics for the two BNP datasets are as follows: 0ng/mL minimum for both datasets, 25th percentile of 75ng/mL for the non-cardiac etiology dataset and 250ng/mL for the obese HFpEF dataset, respective medians of 190ng/ML and 250ng/mL, 75th percentiles of 475ng/mL and 750ng/mL, and maximums of 1075ng/mL and 5000ng/mL.

These statistics were used as constraints to generate a simulated distribution of BNP for each of the cohorts, matching the number of subjects in the non-elevated and elevated LVEDP groups in the present work (258 subjects in the non-cardiac etiology group and 79 subjects in the obese HFpEF group). The performance of BNP was then assessed using this simulated data, from which AUC can be calculated, and thresholds of 50pg/mL and 150pg/mL were applied to calculate any statistics requiring a binary result (i.e., test-negative or test-positive to calculate sensitivity, specificity, PPV, NPV, likelihood ratios). The simulation was repeated for

1000 iterations with the values of the performance statistics averaged and confidence intervals calculated using the distribution of the statistics over the iterations (i.e., values at 2.5th and 97.5th percentiles). The simulation result was analyzed using a Bayesian methodology and to calculate the NRI for the ML predictor based on published methods [34].

## Supporting information

**S1 File. Multivariate clinical predictors of elevated LVEDP.**
(DOCX)

**S2 File. Feature permutation importance.**
(DOCX)

**S3 File. Features.**
(DOCX)

**S4 File. 2x2 contingency Tables & additional statistics.**
(DOCX)

**S5 File. Propensity matching.**
(DOCX)

**S6 File. STARD checklist.**
(DOCX)

**S7 File. Enrolling healthcare centers.**
(DOCX)

**S8 File. Inclusion and exclusion criteria.**
(DOCX)

**S9 File. Placement of the OVG electrodes and PPG sensor.**
(DOCX)

**S10 File. Model specifications.**
(DOCX)

**S11 File. Feature values.**
(XLSX)

## Acknowledgments

The authors are grateful to all patients who participated as study subjects; without you, this work would not have been possible.

## Author Contributions

**Conceptualization:** Sanjeev P. Bhavnani, Rola Khedraki, Travis J. Cohoon.

**Data curation:** Horace R. Gillins.

**Formal analysis:** Sanjeev P. Bhavnani, Horace R. Gillins, Ian Shadforth, Emmanuel Lange, Timothy Burton, Ali Khosousi.

**Funding acquisition:** Sanjeev P. Bhavnani, William E. Sanders, Jr.

**Investigation:** Sanjeev P. Bhavnani, Rola Khedraki, Travis J. Cohoon, Frederick J. Meine, III, Thomas D. Stuckey, Thomas McMinn, Jeremiah P. Depta, Brett Bennett, Thomas McGarry,

William Carroll, David Suh, John A. Steuter, Michael Roberts, Emmanuel Lange, Abhinav Doomra, Mohammad Firouzi, Farhad Fathieh, Timothy Burton, Ali Khosousi, Shyam Ramchandani.

**Methodology:** Sanjeev P. Bhavnani, Rola Khedraki, Travis J. Cohoon, Shyam Ramchandani, William E. Sanders, Jr.

**Resources:** Frederick J. Meine, III, Thomas D. Stuckey, Thomas McMinn, Jeremiah P. Depta, Brett Bennett, Thomas McGarry, William Carroll, David Suh, John A. Steuter, Michael Roberts.

**Software:** Ian Shadforth, Emmanuel Lange, Abhinav Doomra, Mohammad Firouzi, Farhad Fathieh, Timothy Burton, Ali Khosousi.

**Supervision:** Sanjeev P. Bhavnani, Shyam Ramchandani, William E. Sanders, Jr.

**Visualization:** Ian Shadforth, Emmanuel Lange, Timothy Burton.

**Writing – original draft:** Sanjeev P. Bhavnani, Rola Khedraki, Timothy Burton, Shyam Ramchandani, Frank Smart.

**Writing – review & editing:** Sanjeev P. Bhavnani, Rola Khedraki, Travis J. Cohoon, Frederick J. Meine, III, Thomas D. Stuckey, Thomas McMinn, Jeremiah P. Depta, Brett Bennett, Thomas McGarry, William Carroll, David Suh, John A. Steuter, Michael Roberts, Horace R. Gillins, Ian Shadforth, Emmanuel Lange, Abhinav Doomra, Mohammad Firouzi, Farhad Fathieh, Timothy Burton, Ali Khosousi, Shyam Ramchandani, William E. Sanders, Jr.

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
