## [Decision Letter · Decision Letter 0]

21 Jun 2022

PONE-D-22-07622Multicenter validation of a machine learning phase space electro-mechanical pulse wave analysis to predict elevated left ventricular end diastolic pressure at the point-of-carePLOS ONE

Dear Dr. Burton

Thank you for submitting your manuscript to PLOS ONE. After careful consideration, we feel that it has merit but does not fully meet PLOS ONE’s publication criteria as it currently stands. Therefore, we invite you to submit a revised version of the manuscript that addresses the points raised during the review process.

Kindly consider the comments made by the comments of the reviewers and respond accordingly.  Please ensure that your decision is justified on PLOS ONE’s publication criteria and not, for example, on novelty or perceived impact.

We look forward to receiving your revised manuscript.

Kind regards,

Chim C. Lang

Academic Editor

PLOS ONE

Journal Requirements:

“I have read the journal's policy and authors of this manuscript have the following competing interests.

Sanjeev Bhavnani MD is a scientific advisor to Corvista Health and Blumio; consultant to Bristol Meyers Squibb, Pfizer, and Infineon Semiconductor; data safety monitoring board chair at Proteus Digital; has received research support from Scripps Clinic and the Qualcomm Foundation and is member of the healthcare innovation advisory boards at the American College of Cardiology, American Society of Echocardiography, and BIOCOM (all non-profit institutions with all positions voluntary).

Jeremiah P. Depta MD reports consulting fees from Edwards Lifesciences LLC, Boston Scientific, V wave Medical Ltd and Abbot.

Brett Bennett MD reports payment or honoraria for lecture from Philips.

Horace R. Gillins BS, Ian Shadforth EngD, Emmanuel Lange, Abhinav Doomra MScAC, Mohammad Firouzi MSc, Farhad Fathieh PhD, Timothy Burton BComp, Ali Khosousi PhD, Shyam Ramchandami PhD and William E. Sanders Jr. MD report employment by CorVista Health, and stock options in the same.

Frank Smart MD reports grants or contracts from Abbot (GUIDE HF clinical trial), NIH / Ohio State (DCM genetic study), Duke Clinical Research (Transform HF), CorVista Health (Pulmonary Hypertension clinical trial), and participation on a Data Safety Monitoring Board or Advisory Board (Abbott Medical; GUIDE-HF Steering committee).

All other authors report no disclosures.”

5. We note that Supplementary Appendix Section 9 includes an image of a patient in the study.

Additional Editor Comments:

This paper has been reviewed. A number of comments have been raised that require responses and revision.

Reviewers' comments:

Reviewer's Responses to Questions

**Comments to the Author**

1. Is the manuscript technically sound, and do the data support the conclusions?

Reviewer #1: Yes

Reviewer #2: Partly

2. Has the statistical analysis been performed appropriately and rigorously? 

Reviewer #1: Yes

Reviewer #2: Yes

3. Have the authors made all data underlying the findings in their manuscript fully available?

Reviewer #1: Yes

Reviewer #2: No

4. Is the manuscript presented in an intelligible fashion and written in standard English?

Reviewer #1: Yes

Reviewer #2: Yes

5. Review Comments to the Author

Reviewer #1: In this manuscript, the authors aimed to verify if phase space ML analysis with orthogonal voltage gradient and photoplethysmography as features can be used to predict elevated left ventricular end diastolic pressure. Overall, the manuscript is well-structured and clearly written. Below I list some comments for this manuscript.

1. The authors can provide more clear description for the definition of the indices used as features (perhaps provide some figure illustrations).

2. Minor comment: since this is a multi-center study, the authors can provide descriptions about the chosen centers. For example, why choosing these centers? Any special subject characteristic for each center?

Reviewer #2: Introduction:

1. It would be nice to include some literature reviews on any previous studies predicting LVEDP using non-invasive measurement, if any.

Results:

1. Are all the results shown in the results section using the ensembled model?

Discussion:

1. The discussion should focus on the results presented in this study, and the literature reviews should be moved to the introduction.

2. In the limitation, the authors mentioned that study subjects taking diuretics might affect the specificity. Have the authors tried to remove those subjects and rerun the model to test that hypothesis?

3. The authors should add the study populations are patients likely to have CAD to the limitation.

4. The authors should also add the possibility of overfitting to the limitation unless they can justify the large number of features they used in the study.

Methods:

1. The study population are patients that are likely to have CAD, which seems to be a biased dataset to predict LVEDP. If the aim is being able to predict the LVEDP values for just patients who are likely to have CAD, then this would be acceptable, but if the aim is to predict LVEDP for a wider range of populations, then it would make sense to include other patients irrelevant to having CAD.

2. Line 336: Is it without CVD or CAD? If it is CVD, then which diseases are considered here? As CVD is a very wide category.

3. Is the validation group also patients likely to have CAD?

4. What type of catheter was used to measure LVEDP, specs?

5. Line 353: it says the following 4 categories, but there are 5 listed

6. Line 368: what are the symptoms are considered here for HF

7. From Line 414 – 420, please give some specific numbers for the cut-offs, e.g. high-frequency noise above XXX Hz?? Maximum measurable value XXX?

8. Could you give a bit more details on how the features are calculated? I imagine you have both OVG and PPG signals with multiple cardiac cycles. Do you calculate the features cycle by cycle? If so, how do you get the final subject level features based on the cycle features? Do you calculate the features over the whole recorded signal? If so, what is the time window you use for each recording? If the time window is constant, then how many seconds? If not, please justify.

9. Could you provide a list of features used in the appendix? And how they are calculated?

10. Is the outlier detection for the cycle level features or in-between subjects as well?

11. The rationale for choosing the 13 machine-learning models is unclear and how the hyper-parameters were chosen is also unclear. The authors claim that the ensemble outputs are intended to outperform any single model, but would they be able to provide some results showing that?

12. In terms of training the model, what type of training was used? Cross-validation?

13. Can the author provide some information about what they did to prevent overfitting the models?

14. Can the author provide some details on how the 95% CIs were calculated?

General question:

1. How do you avoid overfitting the models?

2. Have you investigated the correlations between features?

Figures:

Figure 8: could you add what each colour and lines represent?

Figure 9: This figure is not very clear. Personally, I don’t think it is necessary to have a figure to describe the random forest, as it is a very well known method. I think it is best to describe what setup was used in the Random Forest, such as bootstrapping=True, how many estimators, any limits for the number of trees, leaves etc. Same for the figure of XGBoost in the appendix.

6. PLOS authors have the option to publish the peer review history of their article (what does this mean?). If published, this will include your full peer review and any attached files.

Reviewer #1: No

Reviewer #2: No

---

## [Author Response · Author response to Decision Letter 0]

24 Aug 2022

We kindly request that you review the cover letter and response to reviewers documents (editor and reviewers, respectively), as they contain images and text formatting intended to facilitate your review, which are not possible to enter into this plain text field.

---

## [Decision Letter · Decision Letter 1]

12 Sep 2022

PONE-D-22-07622R1Multicenter validation of a machine learning phase space electro-mechanical pulse wave analysis to predict elevated left ventricular end diastolic pressure at the point-of-carePLOS ONE

Dear Sir,

Thank you for submitting your manuscript to PLOS ONE. After careful consideration, we feel that it has merit but does not fully meet PLOS ONE’s publication criteria as it currently stands. Therefore, we invite you to submit a revised version of the manuscript that addresses the points raised during the review process.

Kindly note the comments of the reviewer to assist in the further revision needed for your manuscript, Please ensure that your decision is justified on PLOS ONE’s publication criteria and not, for example, on novelty or perceived impact.

We look forward to receiving your revised manuscript.

Kind regards,

Chim C. Lang

Academic Editor

PLOS ONE

Journal Requirements:

Additional Editor Comments (if provided):

Kindly note the comments of the Reviewer to assist in your resubmission

Reviewers' comments:

Reviewer's Responses to Questions

**Comments to the Author**

1. If the authors have adequately addressed your comments raised in a previous round of review and you feel that this manuscript is now acceptable for publication, you may indicate that here to bypass the “Comments to the Author” section, enter your conflict of interest statement in the “Confidential to Editor” section, and submit your "Accept" recommendation.

Reviewer #1: All comments have been addressed

Reviewer #2: (No Response)

2. Is the manuscript technically sound, and do the data support the conclusions?

Reviewer #1: Yes

Reviewer #2: Partly

3. Has the statistical analysis been performed appropriately and rigorously? 

Reviewer #1: Yes

Reviewer #2: Yes

4. Have the authors made all data underlying the findings in their manuscript fully available?

Reviewer #1: Yes

Reviewer #2: No

5. Is the manuscript presented in an intelligible fashion and written in standard English?

Reviewer #1: Yes

Reviewer #2: Yes

6. Review Comments to the Author

Reviewer #1: (No Response)

Reviewer #2: 1. About overfitting

a. Generally, a good ratio between sample size and parameters to avoid overfitting is somewhere above 10 to 1. This means for an average of 149 features, you would need about 1490 subjects. If I am not mistaken, the maximum number included in this study is 684 post exclusions. It is true that the tree-based methods (Random Forest and XGB) and the elastic net can help reduce the risk of overfitting, but it does not mean you will not overfit because you are using these models. I have seen many overfitting cases when people use those models. I do see the authors trying to manage overfitting by limiting the maximum depth, but there are other parameters as well that need to be looked into.

b. One of my additional questions here is how did you choose the hyperparameters for your tree-based models? What type of tunning was performed?

c. Similarly, what type of tunning was conducted for the elastic net model? What makes you choose 0.003 or 0.01 for alpha?

d. Also, an AUC of 0.81 from the validation does not prove that overfitting did not occur since it is just one data point. One can easily argue if there is no overfitting, you could have an AUC of 0.95.

e. In summary, it is still not very clear how the hyperparameters were chosen for each model. And it is unfair for the authors to conclude that the overfitting did not occur based on an AUC of 0.81 from validation. Also, please remove the word “high-performing”, as a word like “high” or “low” is subjective and not scientific.

2. Regarding the type of catheter, the authors have replied to Comment #10 that the choice of the catheter was left to the discretion of each cardiologist, which means the device for measuring the ground truth can be different between time and the medical centre. Have the authors analysed the potential impact of this?

3. Is there a reason that the authors can only provide a list of the families of features used in the study instead of a list containing the actual features?

4. What type of cross-validation did the authors use? Can they specify it in the paper?

7. PLOS authors have the option to publish the peer review history of their article (what does this mean?). If published, this will include your full peer review and any attached files.

Reviewer #1: No

Reviewer #2: No

---

## [Author Response · Author response to Decision Letter 1]

30 Sep 2022

Dear Reviewer #2, 

*Please find these responses in a separate submission file that has the tables formatted for better readability than when copied here*

Thank you for your thorough review and valuable comments. We appreciate and agree with your concern on overfitting and the corresponding risks. We view the recognition of this problem as one of the most significant in the field of machine learning. While we cannot definitively prove that overfitting has not occurred, we believe that the application of our various methods acts to sufficiently mitigate the of likelihood of its occurrence. 

Specifically, our most critical control for overfitting was the single evaluation on a test dataset separated from the development set by enrollment date. Further, the probability of our model achieving the performance of 0.81 by chance on a dataset of that size (which could be the case if our development set was overfit) is low, as shown by the comparison between the curve and the random predictor that lies along the diagonal of the ROC space (p<2E-16 by DeLong’s test). 

We also implemented a series of additional controls to increase the chance that our algorithm would generalize to an unseen dataset: 

 Use of modelling methodologies with regularization and penalty terms that are well-suited to high-dimensional spaces. 

 Cross-validation model development with careful monitoring of fold-to-fold standard deviations

 Careful choice of models to ensemble based on hyperparameter stability

Below we provided detailed explanations of our methodologies to address your specific concerns, and we hope that we have addressed them to your satisfaction. Our responses appear below in bold, provided in italics where applicable in the response, and modified manuscript text in track changes.

Sincerely, 

Sanjeev Bhavnani MD

Specific Comment #1: In order to clarify our imposed safeguards in reference to overfitting, we have revised the text on lines 350-355.

Author Response: The use of an ensemble, as is the case, does not increase the likelihood of overfitting but rather mitigates it by reducing the dependence on a single model. The methods employed to avoid overfitting include the use of cross-validation within the development data, using simple models with regularization and penalty terms, and testing the performance of the model on unseen data (only doing so only once). With respect to the models in particular; first, each model is exposed to an average of 149 features (with a range of 89-194, supplement section 10).

 Generally, a good ratio between sample size and parameters to avoid overfitting is somewhere above 10 to 1. This means for an average of 149 features, you would need about 1490 subjects. If I am not mistaken, the maximum number included in this study is 684 post exclusions. It is true that the tree-based methods (Random Forest and XGB) and the elastic net can help reduce the risk of overfitting, but it does not mean you will not overfit because you are using these models. I have seen many overfitting cases when people use those models. I do see the authors trying to manage overfitting by limiting the maximum depth, but there are other parameters as well that need to be looked into.

Author Response: To clarify the number of subjects – the N=684 that you referenced is the validation cohort, while the models were developed on the N=1272 development cohort (and specifically, each model using a subset of the available cohort, as shown in supplement section 10). With respect to your other comments in this section, we hope that they will be addressed in the responses to your other questions. 

 One of my additional questions here is how did you choose the hyperparameters for your tree-based models? What type of tunning was performed?

Author Response: The possible hyperparameter values are shown below for XGB and Random Forest and were chosen to provide a range over the conservative values of each hyperparameter, with some exponential changes where appropriate to ensure coverage. Each combination of hyperparameters settings was trained within the development dataset only using cross-validation (recall that the ensemble was locked, and the validation data evaluated only once).

XGB Hyperparameters

Hyperparameter Possible Values

Learning Rate 0.1, 0.3, 0.5

Maximum Tree Depth 2, 3, 5, 7

Minimum Child Weight 1, 3, 5

Number of Trees 5, 10, 50, 100

Regularization Alpha 0.1, 0.3, 0.5

Random Forest Hyperparameters

Hyperparameter Possible Values

Maximum Tree Depth 2, 3, 5, 7

Minimum Samples Per Leaf 1, 25, 50, 100

Number of Trees 5, 10, 50, 100, 500

A heuristic stability assessment was performed during the selection of each model. Specifically, for a given model, the cross-validation performance of each “neighboring” hyperparameter specification was compared with that of the model under consideration, and the difference evaluated, with the goal of determining whether that region of the hyperparameter space generated similar cross-validation performances as a measure of stability. To illustrate the method, consider the toy example below using XGB, which assumes that XBG only requires the learning and minimum child weight hyperparameters. Varying these parameters each generates an AUC value, as described in the following table. 

 Minimum Child Weight

Learning Rate 1 3 5

0.1 AUC 0.1,1 AUC 0.1,3 AUC 0.1,5

0.3 AUC 0.3,1 AUC 0.3,3 AUC 0.3,5

0.5 AUC 0.5,1 AUC 0.5,3 AUC 0.5,5

The resultant AUC values can be used to assess the stability of learning rate = 0.3 and minimum child weight = 3 in the following manner. Should the stability value be close to zero, then that indicates that the performance at that specific learning rate and minimum child weight is relatively insensitive to changes to adjacent hyperparameter values. 

Stability= 〖AUC〗_0.3,3- 1/8 (〖AUC〗_0.1,1+〖AUC〗_0.3,1+ 〖AUC〗_0.5,1+ 〖AUC〗_0.1,3+ 〖AUC〗_0.5,3 〖+ AUC〗_0.1,5+ 〖AUC〗_0.3,5+ 〖AUC〗_0.5,5 )

The extension of the methodology is straightforward to higher dimensions of hyperparameters, where the configuration of interest is compared to each of its neighbors in that high-dimensional space, which are found based on changing a single hyperparameter by one step in the assessed range of that hyperparameter. 

Finally, please note that this is a heuristic that we used to guide our selection of the hyperparameters settings, and we do not claim that it is an optimal strategy. For instance, a limitation of this heuristic is the uneven spacing between hyperparameter settings – though we viewed this as necessary to explore the hyperparameter space, it does result in nonuniform distance between neighbors. A possible future extension of this method could correct for that nonuniform distance. 

 Similarly, what type of tunning was conducted for the elastic net model? What makes you choose 0.003 or 0.01 for alpha?

Author Response: Please see the explored hyperparameter space for Elastic Net in the table below. The same hyperparameter stability assessment performed for Random Forest and XBG (as described above in b.) was also performed for Elastic Net. 

Elastic Net Hyperparameters

Hyperparameter Possible Values

Alpha 0, 0.003, 0.01, 0.1, 0.5

Fit Intercept True, False

L1 Ratio 0, 0.01, 0.1, 0.5

Normalize True, False

 Also, an AUC of 0.81 from the validation does not prove that overfitting did not occur since it is just one data point. One can easily argue if there is no overfitting, you could have an AUC of 0.95.

Author Response: The point is well taken. Please see discuss on page one regarding AUC. 

 In summary, it is still not very clear how the hyperparameters were chosen for each model. And it is unfair for the authors to conclude that the overfitting did not occur based on an AUC of 0.81 from validation. Also, please remove the word “high-performing”, as a word like “high” or “low” is subjective and not scientific.

Author Response: We agree that the terms inferring “high/low” are subjective and the text has been altered accordingly on lines 368-369, as shown below. With respect to the hyperparameter selection, we hope that it is now clear from the response to question 1b. 

Finally, the ultimate test of overfitting is the performance on unseen blinded data, which yielded an AUC of 0.81.

Specific Comment #2: Regarding the type of catheter, the authors have replied to Comment #10 that the choice of the catheter was left to the discretion of each cardiologist, which means the device for measuring the ground truth can be different between time and the medical centre. Have the authors analysed the potential impact of this?

Author Response: The reviewer makes an excellent point that variations in catheter technique could affect the consistency of measurements. Such sources of variability are an important concern. However, since 6-French to 8-French catheters are the primary options for left ventricular catheterization, recorded pressure should vary minimally as a function of catheter selection. Moreover, since a variety of catheter types and sizes are used in clinical practice, specifying a particular catheter type may impair the generalizability of the resultant algorithm. 

Specific Comment #3: Is there a reason that the authors can only provide a list of the families of features used in the study instead of a list containing the actual features?

Author Response: Describing all of the features was out of scope for this paper, and therefore we provided feature description at the level of the family in the interest of brevity. However, we did provide detailed information on several of the features (Fig 3 and Supplement Section 3) to allow the reader to gain an appreciation for the prediction mechanism. In addition, many of the features are proprietary in nature. 

Specific Comment #4: What type of cross-validation did the authors use? Can they specify it in the paper?

Author Response: The type of cross-validation used was stratified 5-fold cross-validation repeated for 100 iterations (to vary the train/test folds). Please see the modified text from lines 522-523 below: 

Each of the 13 models were individually performant based on stratified 5-fold cross-validation repeated for 100 iterations (to vary the train/test folds) within the development data (S11) but represent unique analyses of LVEDP assessment.

---

## [Editor Report · Decision Letter 2]

25 Oct 2022

Multicenter validation of a machine learning phase space electro-mechanical pulse wave analysis to predict elevated left ventricular end diastolic pressure at the point-of-care

PONE-D-22-07622R2

Dear Dr

We’re pleased to inform you that your manuscript has been judged scientifically suitable for publication and will be formally accepted for publication once it meets all outstanding technical requirements.

Kind regards,

Chim C. Lang

Academic Editor

PLOS ONE
---

## [Editor Report · Acceptance letter]

5 Nov 2022

PONE-D-22-07622R2 

Multicenter validation of a machine learning phase space electro-mechanical pulse wave analysis to predict elevated left ventricular end diastolic pressure at the point-of-care 

Dear Dr. Bhavnani:

I'm pleased to inform you that your manuscript has been deemed suitable for publication in PLOS ONE. Congratulations! Your manuscript is now with our production department. 

Kind regards, 

on behalf of

Dr. Chim C. Lang 

Academic Editor

PLOS ONE